# Arousal state affects perceptual decision-making by modulating hierarchical sensory processing in a large-scale visual system model

Lynn K. A. Sörensen [1,2]*, Sander M. Bohté [3,4,5], Heleen A. Slagter [6,7‡], H. Steven Scholte [1,2‡]*

1 Department of Psychology, University of Amsterdam, Amsterdam, Netherlands, 2 Amsterdam Brain & Cognition (ABC), University of Amsterdam, Amsterdam, Netherlands, 3 Machine Learning Group, Centrum Wiskunde & Informatica, Amsterdam, Netherlands, 4 Swammerdam Institute of Life Sciences (SILS), University of Amsterdam, Amsterdam, Netherlands, 5 Bernoulli Institute, Rijksuniversiteit Groningen, Groningen, Netherlands, 6 Department of Experimental and Applied Psychology, Vrije Universiteit Amsterdam, Amsterdam, Netherlands, 7 Institute of Brain and Behaviour Amsterdam, Vrije Universiteit Amsterdam, Netherlands

‡ These authors are joint senior authors on this work.
* lynn.soerensen@gmail.com (LKAS); h.s.scholte@uva.nl (HSS)

**Data Availability Statement:** All results and code to reproduce these results can be accessed on the Open Science Framework (https://osf.io/hwfvj/).

**Funding:** This work was funded by a Research Talent Grant (406.17.554) from the Dutch

## Abstract

Arousal levels strongly affect task performance. Yet, what arousal level is optimal for a task depends on its difficulty. Easy task performance peaks at higher arousal levels, whereas performance on difficult tasks displays an inverted U-shape relationship with arousal, peaking at medium arousal levels, an observation first made by Yerkes and Dodson in 1908. It is commonly proposed that the noradrenergic locus coeruleus system regulates these effects on performance through a widespread release of noradrenaline resulting in changes of cortical gain. This account, however, does not explain why performance decays with high arousal levels only in difficult, but not in simple tasks. Here, we present a mechanistic model that revisits the Yerkes-Dodson effect from a sensory perspective: a deep convolutional neural network augmented with a global gain mechanism reproduced the same interaction between arousal state and task difficulty in its performance. Investigating this model revealed that global gain states differentially modulated sensory information encoding across the processing hierarchy, which explained their differential effects on performance on simple versus difficult tasks. These findings offer a novel hierarchical sensory processing account of how, and why, arousal state affects task performance.

## Author summary

Over a hundred years ago, it was first observed that the effect of arousal on performance depends on task difficulty: the Yerkes-Dodson effect. Difficult tasks are best solved at intermediate arousal levels, whereas easy tasks benefit from a high arousal state. Current theories on how arousal affects neural processing cannot explain this effect of task difficulty. Here, we implement a key effect of arousal on cortical processing, a change in

Research Council (NWO, https://www.nwo.nl/) awarded to all authors. The funder had no role in study design, data collection and analysis, decision to publish, or preparation of the manuscript.

**Competing interests:** The authors have declared that no competing interests exist.

neuronal gain, in a computational model of visual processing capable of object recognition. Across a series of experiments, we find that our model can reproduce the Yerkes-Dodson effect behaviorally and that this effect can be explained by where in the processing hierarchy different arousal states optimize sensory information encoding.

## Introduction

Cognitive performance is not stable but varies over time and across situations. Arousal state, the overall activation level of the nervous system, is thought to be a key determinant of variability in cognitive performance [1]. As an example, consider a student taking the same difficult test in two situations: in the first situation, it is a practice test. The student is calm. In the second situation, it is the final exam. The student is very nervous. The difference between these situations is that in the latter, the student might be overly aroused, leading him to perform more poorly on the test. Indeed, a large body of work indicates that arousal level is an important determinant of task performance and perceptual decision-making across species [2–10]. Yet, arousal state does not affect performance on any task in the same way ([11], for reviews see [12,13]). Specifically, while performance on easy tasks improves with arousal level, performance on difficult tasks exhibits an inverted U-shape relationship with arousal level, with performance peaking at intermediate arousal levels. Thus, optimal performance occurs at different arousal states as a function of task difficulty, a phenomenon commonly referred to as the Yerkes-Dodson effect. At the neural level, the inverted U-shape relationship has been associated with the functioning of the locus coeruleus (LC), the central release site of noradrenaline (Fig 1A). The phasic noradrenergic response (upon target detection) in particular also follows an inverted U-shape in its responsivity across LC baseline firing levels. Based on this observation, Aston-Jones and Cohen [14] suggested that this change in responsivity brings about the inverted U-shape in performance, thereby optimizing engagement based on task utility. Yet, this influential view does not address why arousal levels differentially impact performance on relatively easy versus difficult tasks. Indeed, the heterogeneity in findings of recent studies suggests a more complex interplay between arousal state, perceptual performance and other factors such as task difficulty: While some studies report enhanced perceptual performance with increasing baseline arousal levels [5,7], others find performance increases with decreasing baseline arousal levels [15] or observe a curvilinear relationship between performance and arousal [2,3,10]. Possibly, differences in arousal states explored and in task difficulty across studies can account for this diversity in findings. Yet, systematically characterizing the relationship between arousal level and task difficulty is hard to achieve experimentally, in part because it is difficult to cover the full range of possible arousal states and task difficulties within one experimental setting. While this might explain the diversity in findings, it also highlights the need for a suitable modelling framework that can encompass factors such as task difficulty.

Deep convolutional neural networks (DCNN) allow us to address this question. In particular, these networks optimized for object recognition not only parallel human performance on some aspects of object recognition [18], but they also feature processing characteristics that bear a remarkable resemblance to the visual ventral stream in primates [19–24]. Both of these aspects make them an attractive modelling framework for testing computational hypotheses about the link between neural processing and behavior [25,26]. Notably, recent studies show that heightened arousal increases the signal-to-noise ratio (SNR) of sensory neurons [2,4,27,28], both by decreasing variability in spontaneous activity and by increasing neuronal responsivity or gain [2,3,27–29], with consequences for perceptual performance [2,3]. This raises the intriguing possibility that arousal may affect task performance by modulating the

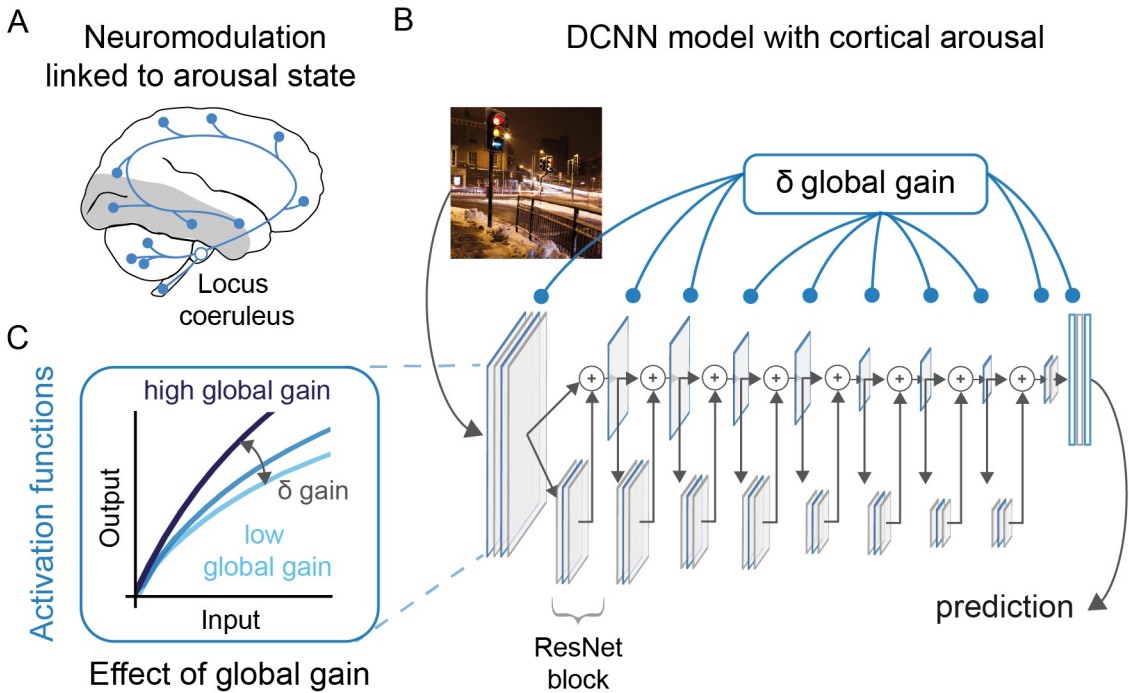

**Fig 1. Global gain as a model for cortical arousal.** (A) Schematic illustration of the projections of the locus coeruleus following [14,16]. The locus coeruleus is located in the pons of the brain stem and features efferent projections throughout the brain. Note that subcortical projections have been omitted here for clarity. The grey shaded areas depict the early visual cortex and the visual ventral stream that we sought to model in this work. (B) Illustration of the DCNN architecture and the locations of the activation functions where global gain, our implementation of cortical arousal, was manipulated (blue framed rectangles). A DCNN takes images as an input and produces a prediction as an output. Importantly, the global gain of the model can be altered with a single parameter that is applied to all activation functions. This takes inspiration from the widespread noradrenergic projections reported for the locus coeruleus (e.g., [14]) and their effects on sensory processing (e.g., [17]). (C) Changes in global gain resulted in a multiplicative scaling of the activation function. Higher gain levels (dark blue) resulted in higher values compared to the baseline (blue), which corresponds to a gain of 1, and reduced gain levels (light blue) lead to lower activation values.

processing of sensory features that perceptual decisions are grounded in. Here, we use DCNNs to test this notion and systematically investigate how a wide range of arousal states modulates sensory processing and perceptual decision-making for tasks of varying difficulty.

In line with the early findings of Yerkes and Dodson [11] and others, we recover the same interaction between arousal state and task difficulty with our DCNN model. That is, also in our model, easier tasks were best solved at high global gain states, whereas difficult tasks were best solved at medium global gain states. This relationship was specific to perceptual difficulty and absent for other types of difficulty (e.g., in response complexity). Moreover, we found that how global gain states affected performance on a given task could be explained by their effects on information encoding across the processing hierarchy. High global gain states enhanced encoding of information in early network features that easy tasks capitalized more on, while intermediate global gain states enhanced encoding of information that was more important for performance on difficult tasks higher in the processing hierarchy. These findings critically inform current debate as to how arousal may impact perceptual decision-making.

## Results

We augmented a DCNN with a global gain mechanism to investigate how arousal state changes in sensory areas may affect performance and relate to the Yerkes-Dodson effect more

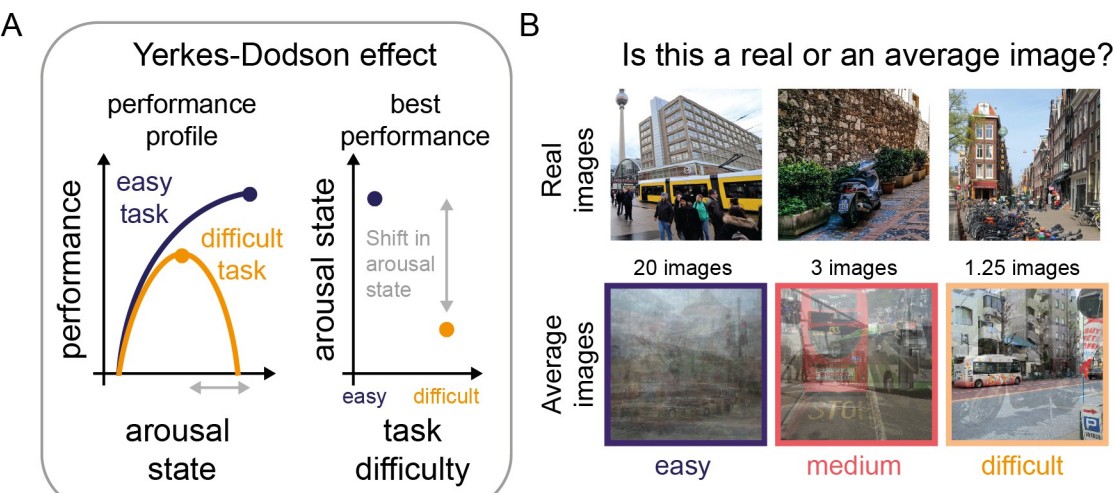

**Fig 2. A perceptual decision-making task to test the Yerkes-Dodson effect.** (A) Left: Schematic of the Yerkes-Dodson effect, showing performance-gain profiles for difficult and easy tasks. While difficult tasks (orange) typically follow an inverted U-shape relationship between arousal state and performance, easier tasks have been associated with an increasing relationship between arousal state and performance. The dots on the performance-gain profiles denote the peak performance. Right: The Yerkes-Dodson effect can be also expressed as a difference in arousal state linked to the peak performance across task difficulty conditions. From this perspective, easy tasks are linked to high arousal states and difficult tasks to medium arousal states for the best performance. (B) Illustration of the network's task. The network was tasked to distinguish between real and average images. To manipulate perceptual difficulty, these average images consisted of an increasing number of images per average image. During training, we fine-tuned an output node for every perceptual difficulty level without applying any global gain changes. This allowed us to dissociate between the network's trained ability and the effects of global gain. For every perceptual difficulty level, we fine-tuned ten network instances. During testing, we presented the fine-tuned networks repeatedly with the same dataset (new set of test images), while adjusting the global gain parameter. This resulted in a binary accuracy result for every global gain value for each of the model instances. Here, we show example images from the real image condition and from the average image condition for three levels of difficulty: difficult (average of 1.25 images), medium (average of 3 images) and easy (average of 20 images). The pictures in B are illustrative examples from the public domain.

specifically. To this end, we used a ResNet18-architecture [30] with a biologically-inspired activation function (see Fig 1C for an illustration, [31,32]). The global gain mechanism targeted all activation functions in the network simultaneously. For every activation function, a change in global gain resulted in a change of response gain ($\delta$ global gain, see Fig 1). All networks were trained in a neutral gain state of 1, corresponding to a standard DCNN without global gain modulation. Subsequently, these trained weights were evaluated across a range of global gain states (without any further training). Using this model allowed us to control global gain state so as to isolate the mechanisms underlying the interaction between arousal state and task difficulty as captured by the Yerkes-Dodson effect.

## A DCNN with global gain replicates the Yerkes-Dodson effect in performance

A hallmark of the effect of arousal state on performance is the interaction between task difficulty and arousal state, the Yerkes-Dodson effect (see Fig 2A for an illustration). In our first set of analyses, we show that our DCNN with a global gain mechanism replicates this effect. That is, we show that easy and difficult tasks require different global gain values for optimal performance across a range of performance measures, with easy tasks yielding the best performance at high global gain states, and difficult tasks performing best at intermediate global gain states.

To study the Yerkes-Dodson effect, one needs to effectively manipulate task difficulty. Task difficulty is commonly manipulated by changing stimulus strength or complexity in perceptual decision-making tasks, for instance, by altering the signal-to-noise ratio of a pure tone

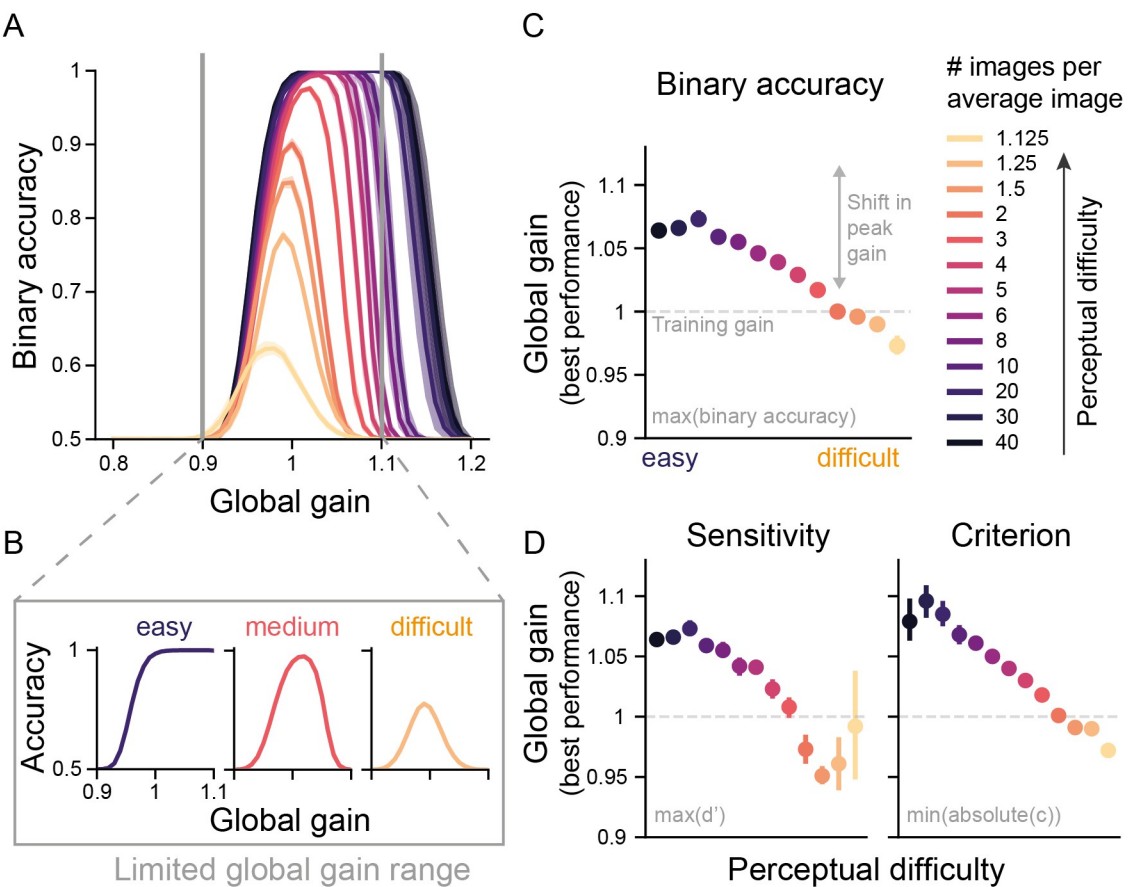

**Fig 3. Incorporating global gain in a DCNN reproduces the Yerkes-Dodson effect.** (A) Binary accuracy for all levels of perceptual difficulty as a function of global gain state. The shaded areas represent the 95% confidence interval (CI) across the ten network instances. (B) As in (A) but for the three levels of difficulty (easy, medium, difficult) and a limited global gain range. It can clearly be seen that as expected, the easy perceptual difficulty condition shows a monotonically increasing relationship between gain and performance, while the difficult perceptual difficulty condition shows an inverted U-shape relationship between gain and performance. Our gain manipulation thus reproduced the Yerkes-Dodson effect in DCNN performance. (C) Global gain level linked to peak performance per difficulty condition in A. As predicted by the Yerkes-Dodson effect, peak performance was associated with reductions in global gain level with increasing task difficulty. The error bars correspond to 95% CI across ten model instances. If multiple gain states were linked to the best performance, the median was used to summarize them. The dashed line corresponds to the neutral gain state from training, during which no global gain changes are applied. (D) As in (C) but now global gain states linked to the performance for sensitivity and bias. Hit and false alarm rates, sensitivity and bias values for all perceptual difficulties and global gain states can be found in S1 Fig.

embedded in a sequence of noise [3] or by phase-scrambling black and white images [7,10]. In a similar vein, we adopted a task, in which the model has to decide whether an image is real or an average image, created by averaging over a variable number of images, taken from a pool of images consisting half of real and half of average images. Task difficulty was manipulated by varying the number of images that were averaged for making the average images. This makes that an average image consisting of few images (e.g., 1.25 images) was relatively similar to and therefore hard to distinguish from a real image and vice versa, an average image consisting of many images was relatively dissimilar and hence easier to distinguish (see Fig 2B for example stimuli). We chose to use averaging as our difficulty manipulation reasoning that it would result in difficult images closer to the training images of the DCNN, thereby increasing the task difficulty. Comparing performance at a neutral gain state (i.e., global gain is 1) in Fig 3A

shows that our manipulation of perceptual difficulty was effective in producing performance differences. For every level of perceptual difficulty, we trained a separate output layer, while keeping the rest of the model weights unchanged. After training, we tested the model on a new set of images across a wide range of global gain states. This allowed us to get high-resolution estimates of the performance-gain profile for every level of perceptual difficulty.

Based on the Yerkes-Dodson effect, we expected to see changing performance-gain profiles across task difficulties and in particular, a shift in the arousal state linked to the best performance (see Fig 2A for our hypothesis). Strikingly, we indeed reproduce the Yerkes-Dodson effect with our global gain manipulation, as reflected in a right-ward shift in the performance-gain profile with a decrease in task difficulty (Fig 3A) that translates to a U-shaped relationship between gain and performance for difficult tasks, but an increasing relationship for easier tasks when a more limited range of arousal is taken into account (see below, Fig 3B). Identifying the global gain level associated with peak performance for every task difficulty (Fig 3C), reveals that there is a quasi-linear negative relationship between task difficulty and the global gain state associated with the best performance, with easy tasks being performed best at high gain states and difficult tasks being performed best at medium gain states. To further dissociate which aspects in the model's performance were changed by the global gain mechanism, we also analyzed performance with regard to changes in sensitivity and bias, two measures from signal detection theory (SDT). Whereas sensitivity refers to the ability to distinguish a signal from noise, bias describes the propensity to answer irrespective of the signal. In turn, accurate performance is defined by high values in sensitivity and a bias close or equal to zero. Following this definition, we identified the global gain state linked to the best performance for both measures. As with accuracy, we observe for both sensitivity and bias that the best performance for easy tasks is observed at high gain states, whereas for difficult tasks it is obtained at medium gain states close to the neutral gain (Fig 3D). These findings show that global gain did not merely result in a change in the networks responsivity or bias, but rather also changed its sensitivity, or its ability to distinguish between real and average images across different task difficulties.

One could argue that the observed inverted U relationship between gain and performance, centered on the neutral gain value of 1, simply reflects state-dependent learning, as the models were trained and fine-tuned in a neutral gain state of 1. However, arguing against this alternative account, the observed pattern of results in Fig 3A shows that the models not only generalized beyond their trained global gain state by achieving similar levels of performance, but critically, that model performance at easier tasks was even better at higher global gain states. This not only displays the models' robustness towards global gain changes, but also suggests a computational advantage of high global gain states for easier tasks.

It is hard to assess a wide range of arousal states experimentally, not only because of experimental constraints, but also because organisms may be very unlikely to visit these extreme arousal states due to homeostatic constraints (hypo- and hyperarousal). Thus, the experimental data so far available likely stem from a much narrower range of arousal states than we explored here. Indeed, it has been argued that some studies might have observed a linear increase in performance as a function of arousal state, because they only sampled from the left-side of the inverted U-shape [33,34]. Mimicking a limited global gain range with our results (Fig 3B) reveals a qualitative correspondence to the experimental data observed for arousal state manipulations across task difficulties [11] and shows how plausibly an increasing relationship between arousal state and performance would emerge for easy tasks, if only low and medium arousal states are experimentally sampled. For our data, inspecting a wider range of arousal states clarifies that, in fact, all performance gain profiles could be readily described as an inverted U (see Fig 3A). This finding can explain how a diverse set of curvilinear relationship can be identified experimentally if the highest arousal states are not accessible.

To summarize these findings, we have replicated the behavioral signatures associated with the Yerkes-Dodson effect across three performance measures. Further, our data offer an explanation for the diversity in experimental findings showing how these could result from sampling a limited range of arousal states covered here by our global gain manipulation.

## Yerkes-Dodson effect is robust to the shape of the activation function

As mentioned above, we here adopted a biologically-inspired activation function in our global gain model. However, most DCNNs feature a rectified linear unit (ReLU, [35]). Since global gain directly acts on the activation functions, the shape of this function might deeply matter for the observed effects on the network's performance. For instance, global gain could simply compress the activation at high levels of activation due to the shape of the activation function (see Fig 3C for an illustration), which cannot happen with a ReLU, since it is fully linear in the positive range (see S2A Fig for a comparison). To determine the specificity of our findings, we repeated the same experiments as presented in Fig 3, but with a DCNN featuring ReLU units. Crucially, we fully replicated our findings (see S2 Fig). This indicates that the Yerkes-Dodson effect observed here is not a by-product of applying global gain to a specific activation function, but rather reflects a global gain mechanism acting on a sensory processing hierarchy of learned perceptual features.

## Yerkes-Dodson effect is absent for another manipulation of task difficulty

Our results so far show that the Yerkes-Dodson effect can be replicated in our model with a manipulation of perceptual difficulty. Yet, in many frameworks, the Yerkes-Dodson effect is linked to task difficulty more broadly [12–14,36,37]. In turn, an obvious question to ask is whether our observations also hold for other manipulations of task difficulty. With task difficulty manipulations, we here simply refer to changes to the task that result in performance changes at a neutral global gain state (i.e., 1, without applying any global gain changes). In our next analysis, we show that another task difficulty manipulation, response complexity, does not reproduce a shift towards higher gain states with decreases in task difficulty, suggesting that the Yerkes-Dodson effect may be specific for perceptual difficulty in our model.

To introduce a new task difficulty manipulation, we altered the number of answer options available to the model (Fig 4B), while keeping all other factors constant such as perceptual difficulty. We applied this manipulation in the context of two different object recognition datasets. Both datasets were curated to be challenging visual search tasks by reducing the amount of informative background differences across categories (for further information see [32]). As can be seen in Fig 4C, this manipulation resulted in differences for the baseline networks (with global gain set to 1), as indexed by AUC, thus showing that the manipulation was effective. This was true for both datasets that we applied this manipulation to (food and street scene images; Fig 4A). If this manipulation also produces an interaction between task difficulty and global gain state, one expects the global gain states associated with the best model performance to differ between the conditions with different numbers of answer options. Fig 4D shows that this is not the case, instead for both data sets, peak performance occurs at the same, medium global gain state independent of the number of answer options. Thus, manipulating the number of answer options for the network merely scaled the gain-performance profile up- or downwards without qualitatively shifting the optimal gain state.

These results also corroborate our findings on perceptual difficulty. Since both datasets were curated to be challenging visual search tasks, they required the analysis of complex visual features. Based on this, one would expect the best performance to be observed for medium global gain states (as was observed in Fig 3 for perceptually difficult tasks). This is indeed what we observed across all six conditions (both datasets, all answer options, Fig 4D).

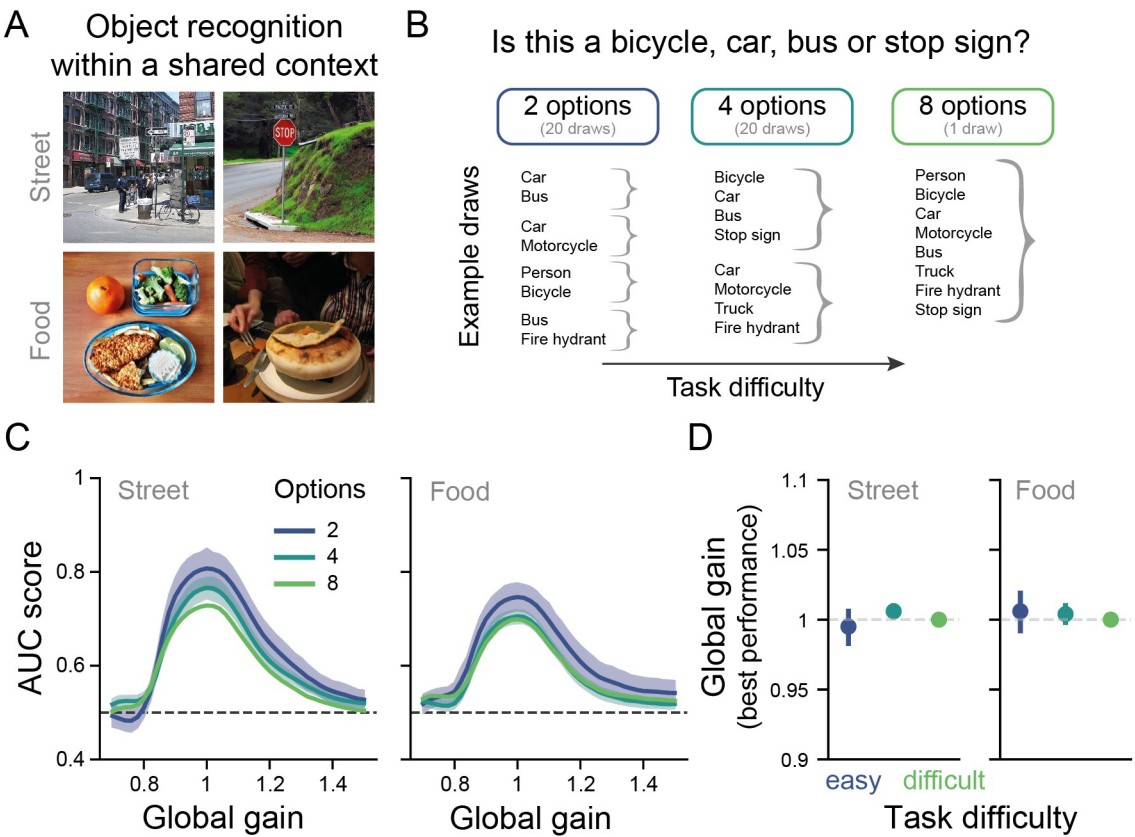

**Fig 4. Assessing the Yerkes-Dodson effect for another kind of task difficulty: Response complexity.** (A) Example images from the two visual search datasets curated from the COCO database [38]. (B) Illustration of how task difficulty was increased by increasing the number of answer options for the street dataset. Easy tasks consisted of only two categories to be distinguished, while difficult tasks entailed choosing from 8 answer options. Different easy tasks were obtained by subsampling from the 8 possible target categories. Task difficulty increased with the number of categories that had to be distinguished. (C) Performance-gain profiles across the three task difficulty conditions for both datasets. The global gain level associated with peak performance was unaffected by task difficulty (number of answer options). Performance was assessed with the area under the curve metric of the receiver operating characteristic curve. The lines represent the mean performance across 20 trained networks for the conditions with 2 and 4 answer options. The shaded areas depict the 95% CI across model instances. The dashed dark grey line depicts chance performance. (D) Global gain state associated with the best model performance was similar for every task difficulty level for both datasets. The dashed light grey line depicts the neutral model, at which no global gain is applied. The example pictures in A are licensed under CC BY 2.0 and were adapted from Flickr [39–42].

In sum, these findings complement our understanding of the Yerkes-Dodson effect in our model in two ways. First, we observed that this finding may be specific to perceptual difficulty in our model, although extending this observation to more tasks in future work is necessary to ascertain this. Second, we have shown that visual search in natural scenes, capitalizing on complex visual features, is best performed at medium global gain states.

## Global gain differentially affects task information across the model hierarchy

The above analyses described how the model's performance, that is, its output, was shaped by changes in global gain states across two tasks. In our next set of analyses, we leveraged the full observability of our model to address how changes in sensory processing may give rise to the Yerkes-Dodson effect. To this end, we first characterized the different stages of processing with regard to this interaction between global gain state and perceptual difficulty. In particular,

we tracked how task-relevant information throughout the network hierarchy was affected by these factors, using a decoding analysis. In brief, this analysis showed that with an increase in model depth there was a shift from higher to medium global gain states for representing the most task-relevant information. Moreover, a second analysis focusing on where in the hierarchy the most task-relevant information was represented for different levels of perceptual difficulty at low, medium and high global gain states revealed that high quality information for solving challenging tasks was specific to late network blocks and that this information in particular was degraded by both low and high global gain states.

To quantify the presence of task-relevant information across model depth (ResNet block; see Fig 1C), we used a linear decoder. Specifically, we adopted logistic regression and predicted whether a model input had been a real or an average image based on the block's activations on a held-out test set (see Methods). Our results show that this can be done almost perfectly for the easier tasks across all model blocks and global gain states (see Fig 5A for examples). Furthermore, it became clear that task difficulty as well as global gain state modulated decoding accuracy across all model blocks (see S3A Fig). In contrast, mean activations did not contain such modulations of task difficulty and were only driven by changes in global gain resulting in an overall increase in activation (see S3B Fig).

To understand how the decoding-gain profiles may be linked to the shift in optimal performance across task difficulty, we performed two complementary analyses. First, we identified the peak global gain state linked to the best decoding performance across model blocks (see Fig 5C and 5D). This analysis indicated how the individual blocks respond to global gain changes depending on task difficulty. Second, we analyzed which model blocks were most informative overall for different levels of perceptual difficulty and global gain (see Fig 5E). This analysis established the relative importance of different model blocks for performing tasks of varying difficulty and showed how their representations were changed by global gain.

The first analysis indicated that, in line with the performance data (Fig 3C), easy task information tended to be best decodable at high gain states, whereas difficult task information was best decoded at medium gain states for most blocks (see Fig 5D). Furthermore, comparing across blocks by aggregating across perceptual difficulties, showed that the global gain state associated with peak decoding was higher for early blocks than later blocks, which exhibited peak decoding at more medium global gain states (Fig 5C). In addition to these general trends, inspecting the global gain states linked to peak decoding performance across task difficulties also indicated an interaction with model depth: Whereas easy tasks showed a negative linear link between model block and peak gain state, this relationship flattened out with increasing perceptual difficulty (see Fig 5D). These findings suggests that later network blocks contained most task-relevant information at medium global gain states, while early network blocks best represented task information at high gain states, in particular for easier tasks.

These results are corroborated by an analysis, which assessed the relative importance of these blocks and showed that in a medium, that is, neutral global gain state (middle panel, Fig 5E), most tasks were best solved relying on information of later network blocks (blocks 5–6). While this also applied to easier tasks, such tasks were also equally well solved based on information of either much earlier (i.e., block 2, circles) or later model blocks (i.e., block 8, diamonds). Thus, information used for solving easy tasks was spread out along the model hierarchy, whereas for challenging tasks, such information was limited to a small subset of blocks. Interestingly, when global gain changes were applied to the network, this pattern mainly changed for the difficult tasks (left and right panel, Fig 5E): for both low and high global gain states, the most informative block was no longer located late in the hierarchy (block 5–6) but instead markedly earlier (block 2 & 3). This suggests that later model blocks were no longer as informative for solving difficult tasks when put in these altered global gain

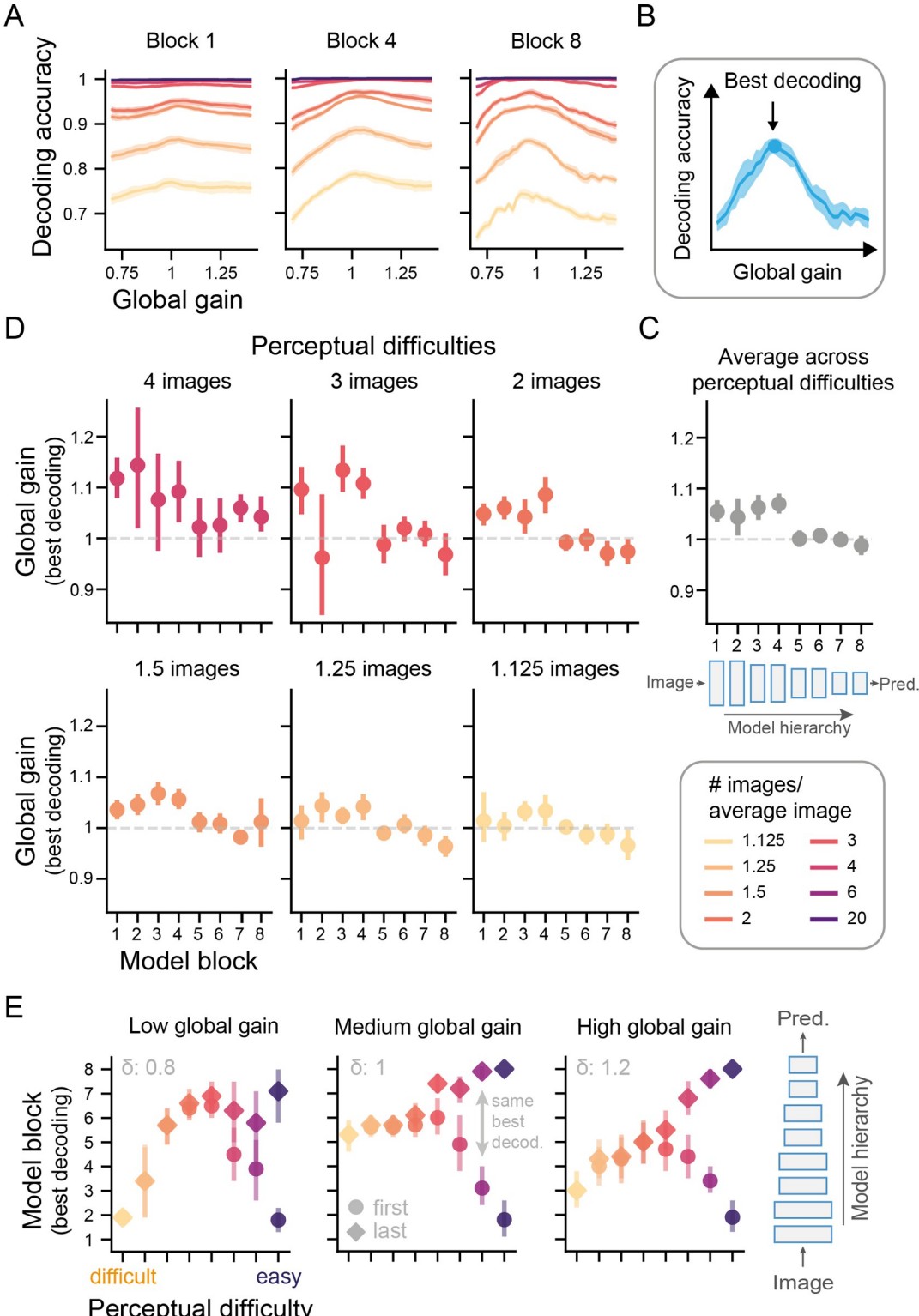

**Fig 5. Global gain differentially affects the representational quality along the model hierarchy.** (A) Mean linear decoding accuracy recorded after the last activation function in a given ResNet block across model instances and shown as a function of perceptual difficulty and global gain (for an overview of all blocks, see S3A Fig). The shaded areas depict the 95% CI across model instances. (B) Analogous to the behavioral analysis, we here identified the global gain state linked to the best decoding

 

accuracy. (C) Model blocks feature most decodable task-relevant information at different global gain states across the hierarchy. The decoding peak refers to the median global gain values linked to the best decoding accuracy across model instances. To obtain an estimate per ResNet block, we averaged these values across perceptual difficulties (shown in D). As in Figs 3C, 3D and 4C, the dotted line refers to the neutral gain states and the error bars describe the 95% CI across model instances. The top part of the figure is a schematic of the DCNN architecture shown in Fig 1 in more detail. Since the easiest tasks (6 and 20 images/average image) were solved almost at ceiling performance across all evaluated blocks and irrespective of global gain state, the peak estimation was unreliable, and we thus excluded it from this analysis (see S3A Fig). (D) The global gain level associated with peak decoding for each model block and perceptual difficulty level separately. This panel shows that easy task information tended to be best decodable at high gain states in early network blocks, whereas difficult task information was best decoded at medium gain states and later model blocks. (E) Global gain states change the location of the most informative model blocks depending on task difficulty. We identified the model block holding the most decodable task-relevant information for every level of task difficulty and global gain state (see S4 Fig for all results). The error bars indicate the 95% confidence interval across model instances. If multiple blocks featured the same best decoding, both the first (circles) and last (diamonds) model blocks are displayed. The link between perceptual difficulty and the most informative model blocks is markedly changed for difficult tasks by global gain changes.

states. In contrast, the easier the task, the less global gain changes appeared to modulate the location of the most informative block along the model's hierarchy.

Taken together, we found that different stages of hierarchical processing were associated with different optimal global gain states for encoding task-relevant information and that the relevance of these stages was strongly modulated by perceptual difficulty. While global gain states facilitated sensory encoding in early network blocks at high gain states, late network blocks contained most task-relevant sensory information at medium gain states, that performance on perceptually difficult tasks more strongly depended on than performance on easy tasks.

## Task difficulty differentially engages model features across the model hierarchy

In the last analysis, we have shown that also decodable information is associated with different optimal global gain states across perceptual difficulties. Crucially, this link varied along the model hierarchy. This observation begs the question whether this linearly decodable information is actually used during network performance. After all, during training, the network learns a set of non-linear transformations on this information and is therefore not limited to linear decoding for performing on this task. To address this question, we devised a causal manipulation that allowed us to selectively reduce the contribution of each model block to the network's performance. In particular, we took advantage of the architecture of our model that has two main branches, one for maintaining and one for processing information (see Fig 6A). By interfering with the information in the processing branch, we were able to inspect the performance-gain profiles with normal and disrupted block function. Like this, we could assess whether feature information in early blocks during higher global gain states is associated with improved performance on easy tasks, as well as whether feature information in later network blocks during medium global gain states benefits model performance on difficult tasks. In brief, we find that indeed early network blocks are the main driver behind optimal performance during high global gain states for easy tasks. Furthermore, this analysis makes it evident that the difficult tasks particularly rely on late network features for performance, which biases them towards medium global gain states in contrast to easy tasks.

Selectively controlling the contribution of a model block to the overall performance is a challenging problem, since the functioning of a network block depends on the activation distributions of the preceding network blocks. To address this, we developed the spatial scrambling method (see Fig 6A). This approach selectively perturbs the information for a given block, but maintains its activation distributions. As a result, this model block's features contribute less effectively to performance and the remaining, unaffected blocks will consequently

 

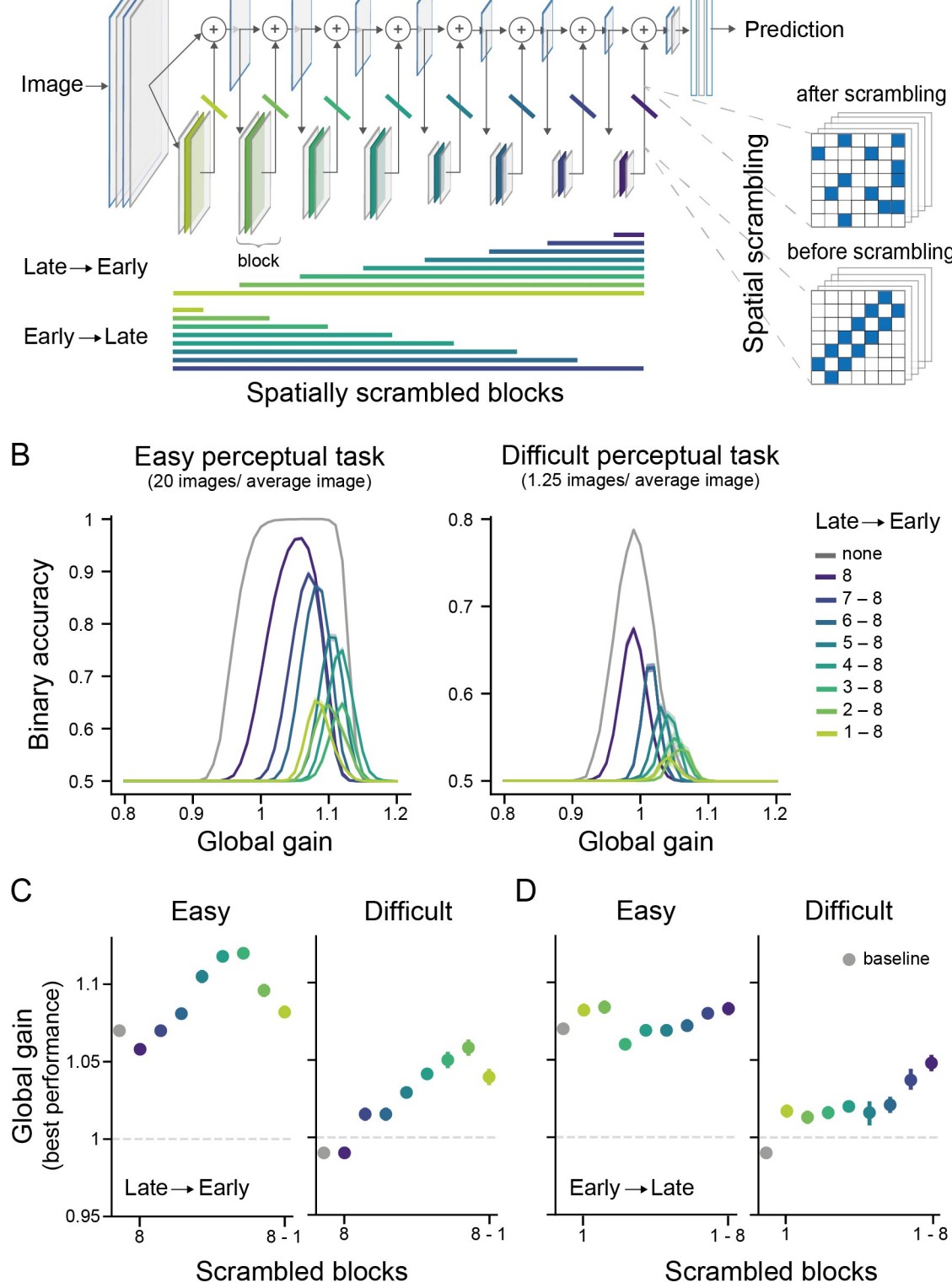

**Fig 6. Task difficulty differentially engages model features across the hierarchy, leading to differences in optimal global gain state in performance.** (A) Spatial scrambling was used to control the contribution of the different model blocks to performance. Spatial scrambling spatially randomizes the activations within a predefined proportion of feature maps in a given model block. This

manipulation was either applied starting from late blocks and then progressing to early model blocks or it was first applied to early blocks and then it progressed to later network blocks. Crucially, after a block is scrambled its informative contribution to the model's performance is strongly reduced (see Methods for an explanation). (B) Starting at later blocks and successively scrambling earlier blocks reveals a pronounced right-ward shift in the performance-gain profiles for both task difficulties. The grey line depicts the performance-gain profile for the baseline network (without any spatial scrambling). Shaded areas show the 95% confidence intervals across repetitions. (C) Determining the global gain states linked to the best performance across different scrambling states of the model shows that in particular the scrambling of later blocks leads to a shift towards higher global gain states for the best performance. The x-axis shows the progression from few to many blocks being scrambled, and in this case, starting from late and progressing to earlier blocks (as is also indicated by the color legend in B). (D) In contrast, first scrambling early blocks had little effect on the global gain state linked to peak performance. This only changed for the difficult task once late blocks were scrambled, which resulted in peak performance being again associated to higher global gain states (as in C). This indicates that while these later network blocks are deployed during recognition, the model operates best overall at medium gain states, since later network blocks operate best at medium global gain states. The x-axis is as in (C) but now progressing from early to later network blocks.

dominate the model's performance (see Methods for an in-depth description of spatial scrambling). Importantly, spatial scrambling led to a relative reweighting of feature contributions to performance. That is, under all circumstances, all network blocks were still affecting performance, but features that were scrambled were less informative and thus exerted less influence on the performance-gain profile. We leveraged this principle to dissect the network's performance, and determine how the features along the model hierarchy differentially contributed to performance of easy and difficult tasks. Specifically, we either increasingly added spatial scrambling from the top to the bottom of the network, thereby making the network more and more reliant on feature information contained in the early blocks or we applied it from the bottom to the top of the network, so that late network features drove performance (see Fig 6A).

Our results so far suggest that high global gain states benefit sensory processing in early blocks. Therefore, we expected that increasingly disrupting late network function (Late to early), i.e., assigning more weight to earlier blocks, would lead to a right-ward shift in the performance-gain profiles. That is, we expected to find the best performance during higher global gain states in this case, since early features are encoded well during high global gain states. Fig 6B shows that this expectation was met for both perceptually difficult and easy tasks. In particular, we observed that greater reliance on processing in early network blocks led to peak performance at higher global gain states (see Fig 5C, *Late to early*). Notably, this resulted in very high global gain states compared to the baseline network (see Fig 6B and 6C). Across both task difficulty conditions, peak performance shifted to the highest global gain states if almost all late network features were scrambled. These findings suggest that higher global gain states may facilitate network performance by enhancing feature content in early network blocks. Beyond this general effect, there was also a difference between task difficulty conditions: As perceptually easy tasks relied more on sensory processing in early network blocks than perceptually difficult tasks, scrambling late network blocks had less of an effect on performance on easy tasks, as shown in Fig 6B.

In the absence of spatial scrambling, we observed that in contrast to easy tasks, the network's performance on difficult tasks was best at medium global gain states (Fig 3). As we found optimal decoding of task-relevant information in later blocks at medium global gain states for difficult tasks (Fig 5), we expected that first scrambling early blocks would not substantially affect the relationship between gain and performance on difficult tasks until scrambling also included later blocks. Here we assumed that in particular these later network features determine the performance-gain profile, since these features are specifically linked to high performance at medium global gain states. To test this, we next scrambled the network blocks increasingly from early to late blocks (see Fig 6C, *Early to late*). Indeed, the global gain state associated with the best performance remained relatively stable and visibly lower compared to the early-to-late condition for both task difficulty levels (see S6 Fig for performance-

gain profiles). Only once also the very last network features were affected (block 7 and 8), there was a notable increase towards higher gain states for peak performance on the difficult task. This finding indicates that in particular information contained in block 6 was important for this task, biasing the performance-gain profile towards lower global gain states. This conclusion is also in line with the decoding result suggesting block 5–6 to hold most task-relevant information for difficult tasks. Taken together, we find that sensory processing in later network blocks is more critical for solving perceptually difficult tasks and that medium global gain states facilitate later processing and hence performance on difficult tasks. In contrast, easy tasks rely more on sensory processing in early blocks, which operate better at higher global gain states.

Taken together, these findings indicate that the effects of arousal states on sensory processing can explain at least in part the effects of arousal state on perceptual performance as a function of task difficulty, commonly referred to as the Yerkes-Dodson effect. More concretely, global gain level may differentially affect computational features across model depth, which, depending on the level of global gain and task difficulty, benefits performance on some tasks, but impairs performance on other tasks.

## Discussion

Since the pioneering work by Yerkes and Dodson [11], a large body of research has demonstrated the prominent role of arousal, not just in sleep and wakefulness, but in task-related neural processing and performance (e.g. [4,12,14,43,44]). Yet, precisely how arousal states affect task performance, and specifically, why high arousal states impair performance on difficult, but not easy tasks, as reflected in the Yerkes-Dodson effect, is still unclear. We here capitalized on recent insights from neuroscience to model the Yerkes-Dodson relationship, exploiting the unique ability of DCNNs to systematically simulate the full range of possible arousal states while performing a task. Traditionally viewed to reflect a modulation of perceptual decision-making by the locus coeruleus [14], we here show that we cannot only reproduce the Yerkes-Dodson effect in our model by augmenting it with a global gain mechanism, but also that sensory effects of arousal states are sufficient to account for the Yerkes-Dodson effect. Investigating the inner workings of our model, we show that different global gain states optimize sensory encoding of task-relevant information at different stages of hierarchical processing thereby in particular affecting performance on tasks that more critically rely on those stages of processing. That is, high global gain states facilitated early model processing, thereby enhancing performance on perceptually easy tasks, while intermediate global gain states facilitated late model processing, enhancing performance on perceptually difficult tasks. Altogether, we provide a new perspective on the long-standing question as to how arousal can facilitate and impair task performance by highlighting the complex interaction between arousal and hierarchical sensory processing.

### DCNNs as promising avenue for modelling the effects of cortical arousal

To our knowledge, this is the first large-scale simulation of the effect of arousal state on visual processing. While other computational accounts have described the effects of gain in small-scale models [45,46], these could not address factors such as perceptual task difficulty. Here we critically extend this work by directly linking global gain to effects on perceptual performance using natural images and a task that varied parametrically in perceptual difficulty. As any model, our model omits some crucial details such as the temporal signature of these effects typically used to delineate phasic from tonic arousal effects in pupillometry (for a discussion, see [47]), oscillatory signatures linked to synchronized and desynchronized states [48] or the

diversity of effects in different cortical target neurons (i.e., [27,49,50]). Yet, our findings provide a starting point for understanding the complex interaction between a spatially unspecific neuromodulator such as noradrenaline and sensory processing. As apparent from our results, working with such a model both affords an in-depth investigation of model processing (Fig 5) and more importantly, allows for directed causal manipulation (Fig 6). Our study highlights how this approach can advance our understanding, both by accounting for perceptual complexity and by leveraging computational mechanisms derived from neural data. Furthermore, our model also introduces state changes into the DCNN modelling framework, thereby contributing to the effort to develop computational models approximating the computations in the ventral visual stream and beyond [51]. While current efforts are mainly directed at describing trial-averaged performance [52] and neural data [23], this approach may overlook both the computational mechanisms and effects of interindividual state changes [1,14,48,53]. For instance, our results reveal how changes in global gain effectively reconfigure computational features and thereby dramatically change model performance. This reweighting of computational features as a function of global gain state may be a strategy that could also subserve visual processing in the brain, benefitting some visual tasks, while hampering others, as we have shown here. However, while there is a growing body of evidence for arousal effects on sensory processing in mice [2,3,17,27,28,43], non-human primates [54,55] and humans [5,7,10], to our knowledge, it is still unclear how arousal state impacts higher visual cortices essential for object recognition and how those effects may be dependent on the visual task at hand. Future studies should also examine how specifically information is differentially propagated throughout the hierarchy as result of global gain changes, for example, by studying the interaction between global gain changes and computational circuits, such as ResNet blocks, in smaller-scale models. They should also extend our findings to other architectures to ensure that our findings generalize beyond ResNet architectures. We hope that our model will serve as an important steppingstone for experimentally and theoretically investigating how global state changes, such as arousal, affect sensory processing and perception.

### Link between the model's global gain states and biological arousal states

We could investigate the effect of arousal state across a wide range of arousal states allowing us to carefully describe the performance-gain profiles associated with different task difficulties. While not all gain values examined here are biologically plausible, it was our goal to also describe the tails of the distributions that are usually not accessible experimentally. A biological system bound to homeostasis likely never visits such extreme arousal states. The decay in performance on easy tasks observed for the highest gain states is therefore likely not commonly observed in biological systems and in practice may more resemble a continuous increase as has been also suggested by others [33,34]. Moreover, using our parametric perceptual task, we did find that the performance-gain profiles observed as a function of perceptual difficulty in fact lie on a continuum (see Fig 3). Thus, one may expect to find a shift in peak performance while varying arousal and perceptual difficulty. Importantly, our data also suggests that most variation across levels of task difficulty is expected to be observed at high arousal states. Future behavioral studies adopting a continuous perceptual difficulty manipulation across a wide range of arousal states will be key in testing our predictions.

### DCNNs with global gain reproduce key behavioral signatures linked to arousal states

In addition to the Yerkes-Dodson effect, our model also replicates a number of other recent findings from studies that assessed arousal state by measuring pupil size before or during

stimulus processing [17,28]. In line with our model, across many studies, the highest perceptual sensitivity has repeatedly been linked to medium arousal states for both visual and auditory tasks in mice and humans [2,3,10,34]. All of these tasks were challenging and linked to the characteristic inverted U-shape across sensitivities. Crucially, this connects our model to a larger framework in which arousal regulates the efficiency of information processing ([4,5,14,54,56] but see [55]).

## DCNNs as a model for visual processing and perceptual difficulty

We here found that the Yerkes-Dodson effect in our model could only be observed when manipulating perceptual difficulty, whereas our model was insensitive to another manipulation of task difficulty (i.e., number of answer options). This finding may seem at odds with perspectives linking the Yerkes-Dodson effect to manipulations of task difficulty more generally (e.g. [12–14,36,37]). Yet, there is long-standing debate about what the terms (emotional) arousal and task difficulty specifically denote, and both have been repeatedly criticized for their vagueness [12,13,37]. Indeed, recent studies on this topic relied mostly on difficulty manipulations in the perceptual domain [3,5,7,10,15]. Our results add to this body of work by suggesting that the interaction between arousal state and task difficulty may be selective to the perceptual domain. However, to firmly conclude this, these observations should be extended to other tasks and difficulty manipulations in future studies.

The Yerkes-Dodson effect was initially observed with a black and white visual discrimination task and task difficulty was manipulated by varying ambient light during a learning task [11,57]. For the current study, we chose a manipulation of task difficulty that mimicked those of more recent studies [2,3,5,7,10], which varied the signal-to-noise ratio (SNR) of target stimuli, with tasks with a higher SNR considered easier. In those studies, the SNR is usually increased by varying the target's intensity, for instance, its contrast. While our approach of averaging a different number of images (see Fig 2) also varies the SNR, we did not vary the images' contrast, and this may be an important avenue for follow-up experiments. In a similar vein, many different stimuli have been used for studying the effects of arousal state on perceptual processing: Some studies also used naturalistic stimuli as targets [5,7,10], as we did here, whereas others used more artificial stimuli such as Gabor patches [5], moving dots [15], drifting square-wave gratings [2] or pure tones [3] as target stimuli. Our observation of an interaction between global gain state, perceptual difficulty, and the model's processing hierarchy predicts that tasks that rely on feature distinctions of varying complexity should also have a different relationship to arousal state. Importantly, the current modelling framework makes it possible to assess the interaction between stimuli, task difficulty and arousal state more systematically in future studies.

With our work we do not wish to suggest that our model can serve as a model for the brain as a whole, since DCNNs largely capture processing in the visual ventral stream [19–24]. Thus, our study should be interpreted as a minimal model for reproducing the Yerkes-Dodson effect, and we cannot exclude that other task factors, which mainly engage other cortical areas, would not produce or contribute to a similar pattern of results. Rather, our findings are a proof of principle of the complex interactions arising from the combination of global neuro-modulatory signals such as noradrenaline and hierarchical sensory processing.

Indeed, another indication for the fact that our model may have specifically captured more initial stages of perceptual decision-making is suggested by our observation of a continuous decrease in criterion with increasing global gain for all task difficulty levels, related to continuous increases in both hit and false alarm rates with increasing gain (S1 Fig). This finding contrasts with a study by McGinley and colleagues [3] in mice that found a decrease in hit rate

during high arousal states and correspondingly, a U-shaped link between the mice's criterion and arousal state. Our model does not reproduce this decrease in hit rate during high arousal states, suggesting that it may not well capture all stages of perceptual decision-making. Being a model for visual processing, it is not designed to mimic processes linked to other aspects of decision-making, such as flexibly criterion setting or behavioral strategy (e.g. [53,58]). Notably, a dissociation between perceptual and other aspects of decision-making is supported by a number of recent studies, which attribute a reduction in choice bias to phasic arousal and computations in the brain stem arousal systems and prefrontal cortices [8,59,60]. Nonetheless, our model provides a promising starting point for also modelling these more complex aspects of decision-making.

## Concluding remarks

To conclude, our results suggest that DCNNs with a global gain mechanism can serve as a computational model of the sensory effects of cortical arousal. We here showcased how such a model can capture the behavioral signatures linked to arousal states across species as well as provide mechanistic insights, thereby providing a new account of the Yerkes-Dodson effect. Our results also illustrate the value of neuroscience-inspired computer vision algorithms in the study of brain-behavior relationships by revealing the complex interplay between global gain effects and hierarchical sensory processing.

## Methods

### Overview

We modelled cortical arousal by augmenting a DCNN with a global gain mechanism (see *Model*) and tested it on two tasks (see *Tasks*). In addition, we also investigated effects on processing by both assessing the model's decodable task information and its activations (see *Block information & activations*) and disrupting its processing (see *Model disruption–Spatial Scrambling*).

### Model

As a base architecture for our DCNN, we adopted a first generation ResNet18 [30] that was trained with a sigmoidal-like activation function [31] on the ImageNet database [61] and achieved a top-1 accuracy of 64.04% on the validation set. Commonly, DCNNs use a rectified linear unit (ReLU) as activation function and due to the use of this alternative activation function, performance was slightly lower than models trained with ReLU (typically around ca. 69%). While we preferred to use a sigmoidal-like transfer function for all the main experiments because of its saturating property (see Fig 1A), akin to biological neurons at extreme values [62], we also replicated the perceptual performance experiments (Fig 2) with a ReLU activation function. For this we used a pretrained ResNet18 model retrieved from [63] with a top-1 accuracy of 68.24%.

### Global gain mechanism

Arousal has been linked to increases in firing rates in early visual cortex in response to sensory stimulation, that is, to increases in sensory gain, (e.g. [27]). These increases in gain were suggested to be a result of a global release of noradrenaline (in conjunction with other neuromodulators [48]), making them spatially unspecific [4,14,16,44,48].

Building on prior modelling work [45,46], we here simulate such global gain increases in our DCNN by scaling all activation functions with a single global gain parameter $\delta$. For the

sigmoidal-like transfer function, this results in:

$$f(S) = \delta \max\left(0, \frac{h}{\exp\left(\frac{c_1 S + c_2}{c_3 S + c_4}\right) - 1} - c_0 + \frac{h}{2}\right),$$

where $f(S)$ describes the outgoing activation and $S$ is the incoming activation. The second part of the equation describes the sigmoidal-like properties of the activation function where the constants $h$, $c_0$, $c_1$, $c_2$, $c_3$ and $c_4$ were derived from a spiking neuron model, capturing the mapping between the incoming current and the average postsynaptic potential over infinite time steps [31]. Note that an activation function according to this formulation is relatively comparable to a ReLU for small values but exhibits large differences for large input values due to its sigmoidal-like saturation (see S2 Fig for a visual comparison).

For a model with ReLU activation function, global gain $\delta$ also simply scales the activation function:

$$f(S) = \delta \max(0, S)$$

For both models, global gain $\delta$ targeted all activation functions in the model, thereby increasing or reducing response gain everywhere simultaneously.

## Tasks

**Parametric perceptual difficulty task.** The Yerkes-Dodson effect—the relationship between arousal level and performance as a function of task difficulty—is commonly studied by manipulating sensory difficulty [3,11,57]. To recreate an analogous situation for our model, we developed a task in which we could parametrically manipulate the visual complexity required to solve a binary discrimination task. This task required the model to distinguish a real image from an average image, an image that was created by averaging over different numbers of scene images. That is, an average image could either be based on only 1.25 images (average of a normal image and another image with an alpha level of 25%), which rendered it very similar to a real image and thus perceptually difficult to distinguish from a real natural scene image, or it could be based on averaging over up to 40 images, making the average image very dissimilar from a natural scene and thus the discrimination task easier (see Fig 2B for examples). The average images were constructed from a separate set of images than those used as real images, thereby avoiding overlap in image features. All images were taken from a dataset consisting of street scenes, described here [32].

For every level of perceptual difficulty, a separate sigmoidal output node of the DCNN was fine-tuned, while the rest of the weights were kept unchanged and the global gain parameter was set to 1. The respective models were fine-tuned for 50 epochs on 6040 training images and validated using 2592 images, each containing 50% natural scenes for 10 network instances. The data shown in Fig 3 for a global gain of 1 show the performance of these models on the held-out test set (2159 images). To construct the performance-gain profiles depicted in Fig 3, the same test set was evaluated using the same fine-tuned DCNNs.

For evaluating the signal detection properties, we calculated $d'$ and the *criterion* as follows:

$$d' = z\left(\frac{hits}{hits + misses}\right) - z\left(\frac{FA}{FA + CR}\right)$$

$$criterion = -0.5\left(z\left(\frac{hits}{hits + misses}\right) + z\left(\frac{FA}{FA + CR}\right)\right),$$

where $z$ corresponds to a z-transform and $FA$ to false alarms and $CR$ to correct rejections.

**Visual search with a varying number of options.**   To assess how the results from the parametric perceptual difficulty task generalized to other scenarios, we also tested the model on a visual search task across two scene contexts in which we varied the number of answer options. The benefit of this was two-fold. First, we could test how our the DCNN responds to another factor of task difficulty, with an eight-way classification being more difficult than a two-way classification. Secondly, we could assess whether we would also observe an inverted U-shaped gain-performance profile for visual search, a perceptually difficult task. We tested the DCNN across two search contexts (street scenes and food scenes). These datasets were curated to be challenging for visual search, while reducing other informative features such as the background (for a detailed description, see [32]).

We fine-tuned an output layer with sigmoid nodes separately for each number of answer options (2, 4, 8 categories), while keeping the remaining model weights unchanged. The models were trained on multi-label images for 75 epochs and tested on single-label images. To reduce the number of answer options, but prevent differences between the different answer option conditions in categories included, we randomly sampled with replacement from the 8 categories over 20 iterations thus providing us with the depicted standard deviation in Fig 4 for the conditions 2 and 4.

## Block information

We examined the effects of global gain on model block information to determine how our manipulation of arousal, at the level of sensory processing, may have affected performance as a function of perceptual difficulty. To assess the amount of task-related information that is linearly decodable, we used a logistic regression classifier. In particular, we used this decoding approach on the activations from the last activation layer in each ResNet block, thus just before the residual and skip connections split. We ran 5 iterations with randomly drawn training and test datasets of 1000 images each for every combination of level of perceptual difficulty and gain value. For the logistic regression, we used the standard parameters as implemented in the scikit-learn package [64].

## Block activation

Next to examining effects of global gain on block information, we also examined gain effects on task-related activation. To obtain a measure of activation along the hierarchy of the model, we recorded the mean activation across all dimensions for the test set used in Fig 3 for varying degrees of global gain and perceptual difficulty for all trained model instances in the same layers as included in the decoding analyses.

## Model disruption—Spatial scrambling

To manipulate the model's processing, we developed a method to selectively reduce the contribution of a given block to the model's performance, named spatial scrambling (Fig 6A). In particular, we leveraged the organizational principle of ResNet blocks, consisting of a residual and a skip branch (see Fig 6A), where the residual branch contains the majority of the additional task-relevant features extracted for a given block (in the convolutional layers). In a functioning ResNet, the features from the residual branch are added back to the skip branch, which did not undergo this extra computational step. The information transmitted via the skip connection is thus identical to the model activations before they entered the residual branch (see Fig 6A). Spatial scrambling interferes with these residual branch features to effectively reduce their contribution to the model's performance. Specifically, we spatially scrambled the feature maps and did so with an increasing proportion of all feature maps. Thereby spatial scrambling

maintained the distributional statistics within a feature map (in contrast to other methods such as Dropout [65] or lesioning), but interfered with the information contained therein.

By increasing the proportion of randomly chosen feature maps affected by spatial scrambling while probing the network for its performance, it is possible to estimate how robust the network is to these manipulations while at a neutral gain of 1. This is illustrated in S5 Fig, which shows how performance drops as a function of features maps being targeted by spatial scrambling (importance curves). The gradual decay of these curves informs about the relative relevance of these model features for performance. For instance, a model's block importance curve that quickly falls off as a function of an increased proportion indicates that this feature was relatively essential to the model's overall performance. For the main experiments in which we also varied global gain, we determined the proportion of randomly chosen feature maps at which a model block dropped to 20% of its possible contribution to network performance (dots in S5 Fig). This ensured that applying spatial scrambling to every targeted ResNet-block had approximately the same relative impact on model performance.

In a next step, we used this approach to dissect which part of the model most strongly drove the performance-gain profiles to be best at medium or heightened global gain states. For this analysis, we only applied the same spatial scrambling rate identified based on the importance curves for every block but changed the global gain of the model. In particular, we were interested to dissociate the relative influence of early versus later network blocks on the performance-gain profiles. To this end, we either gradually disrupted later model blocks first and successively added earlier model blocks or the other way around, thereby reducing the informational value for performance of later or earlier blocks respectively. Doing this for different task difficulties then allowed us to inspect the performance-gain profiles as a function of disrupted early or later network block function. Note that spatially scrambling e.g., earlier blocks likely also affects the information contained at later blocks. Hence, this manipulation provides an indication of the relative importance of early vs. late blocks on performance-gain profiles but does not allow to determine the absolute importance of any block since all blocks simultaneously affect the performance-gain profile. All spatial scrambling experiments were repeated ten times on the same set of training weights.

## Software

All results presented here were obtained while relying on the following Python packages: NumPy [66], keras [67], TensorFlow [68], Pandas [69], Scikit-Learn [64] and SciPy [70]. Data visualization was done using matplotlib [71] and, in particular, seaborn [72].

## Supporting information

**S1 Fig. Signal detection properties as a function of perceptual difficulty and global gain.** Shaded areas depict the 95% confidence interval across ten model instances. The grey vertical lines are in reference to Fig 2D and serve to illustrate the results for a more narrowly sampled global gain range.
(TIF)

**S2 Fig. A DCNN with global gain and a linear activation regime also reproduces the Yerkes-Dodson effect.** (A) Comparison of the activation functions used for the main experiments (sigmoidal) and an alternative (rectified linear unit, ReLU) evaluated in this figure. While activation regimes are comparable for small values, they diverge for large values. (B) Binary accuracy for all levels of perceptual difficulty as a function of global gain state. The shaded areas represent the 95% confidence interval (CI) across ten network instances in B and C. As in

Fig 3, this pattern of results also reproduced the Yerkes-Dodson effect in the DCNN's performance. (C) Signal detection properties as a function of perceptual difficulty and global gain. (D) Global gain level linked to peak performance per difficulty condition in B. In line with the Yerkes-Dodson effect, peak performance was again associated with reductions in global gain level with increasing task difficulty (as in Fig 3). The error bars correspond to 95% CI across ten model instances. If multiple gain states were linked to the best performance, the median was used to summarize them. The dashed line corresponds to the neutral gain state from training, during which no global gain changes are applied. (E) As in (C) but now global gain states linked to the performance for sensitivity and bias.
(TIF)

**S3 Fig. Global gain changes affect task-related information and activation.** All figure conventions are the same as in Fig 5A.
(TIF)

**S4 Fig. Link between most informative network block and global gain state shown separately for different levels of perceptual difficulty.** The vertical dashed lines indicate the data shown in Fig 4E and 4B. The shaded areas show the 95% confidence interval across ten network instances.
(TIF)

**S5 Fig. Importance curves obtained from spatial scrambling at neutral gain states.** The dots represent the scrambling rate at which 20% of the baseline performance is maintained.
(TIF)

**S6 Fig. Performance-gain profiles for scrambling early to late network blocks.** A shows a perceptually difficult task and B a perceptually easy task. All figure conventions are the same as in Fig 6.
(TIF)

## Acknowledgments

The authors would like to thank the members of the Neuro-Computational Vision Journal Club and in particular Iris Groen, Noor Seijdel and Tomas Knapen for their feedback on an earlier version of this manuscript.

## Author Contributions

**Conceptualization:** Lynn K. A. Sörensen, Sander M. Bohté, Heleen A. Slagter, H. Steven Scholte.

**Data curation:** Lynn K. A. Sörensen.

**Formal analysis:** Lynn K. A. Sörensen.

**Funding acquisition:** Heleen A. Slagter, H. Steven Scholte.

**Investigation:** Lynn K. A. Sörensen, Sander M. Bohté, Heleen A. Slagter, H. Steven Scholte.

**Methodology:** Lynn K. A. Sörensen.

**Project administration:** Lynn K. A. Sörensen, Heleen A. Slagter, H. Steven Scholte.

**Resources:** H. Steven Scholte.

**Software:** Lynn K. A. Sörensen.

**Supervision:** Sander M. Bohté, Heleen A. Slagter, H. Steven Scholte.

**Validation:** Lynn K. A. Sörensen.

**Visualization:** Lynn K. A. Sörensen.

**Writing – original draft:** Lynn K. A. Sörensen.

**Writing – review & editing:** Lynn K. A. Sörensen, Sander M. Bohté, Heleen A. Slagter, H. Steven Scholte.

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
