## [Decision Letter · Decision Letter 0]

8 Nov 2021

Dear Mrs. Sörensen,

Thank you very much for submitting your manuscript "Arousal state affects perceptual decision-making by modulating deep hierarchical sensory processing" for consideration at PLOS Computational Biology.

As with all papers reviewed by the journal, your manuscript was reviewed by members of the editorial board and by several independent reviewers. In light of the reviews (below this email), we would like to invite the resubmission of a significantly-revised version that takes into account the reviewers' comments.

We cannot make any decision about publication until we have seen the revised manuscript and your response to the reviewers' comments. Your revised manuscript is also likely to be sent to reviewers for further evaluation.

Sincerely,

Leyla Isik

Associate Editor

PLOS Computational Biology

Thomas Serre

Deputy Editor

PLOS Computational Biology

Reviewer's Responses to Questions

**Comments to the Authors:**

Reviewer #1: The paper covers a large and interesting topic in perceptual sciences, one that is very sensible to investigate with a DCNN so as to connect neural modulation with task performance. I found the manuscript clearly written and presented for the most part. I do, however, have some small clarifications that are needed throughout along with some reining in of claims. I also believe there are significant issues with the scrambling analysis and how the authors have interpreted these results.

I'm concerned the title overstates the results, as it doesn't read as being model-specific. A small change such as "Arousal state can affect perceptual decision-making by modulating deep hierarchical sensory processing" would help.

"Furthermore, stimulus-unrelated neural activity in early visual cortices is closely related to arousal [29]. This raises the intriguing possibility that arousal may affect task performance by modulating the processing of sensory features that perceptual decisions are grounded in." How does it follow that "processing of sensory features that perceptual decisions are grounded in" is what is modulated if it is *stimulus-unrelated* neural activity that is impacted by arousal?

The authors use two different style of tasks (detecting averages and cluttered classification) to claim that the Yerkes-Dodson difficulty effect is based on perceptual difficulty and not response difficulty. The use of only one type of task to represent each style of difficulty is insufficient to support this claim. If the authors want to make this claim they will need to test out other examples of perceptual difficulty and response difficulty. For example, they could use different levels of clutter with the same readout options to determine if clutter level (a form of perceptual difficulty) shows the Yerkes-Dodson effect. As another test of response difficulty, they could also use their averaged images to create a task where the network needs to determine how many images are in the average, with increasingly narrow ranges as options (for example, an easy binary test for greater or less than 5 images, and more difficult tasks to classify 20-40, 0-20, etc). Without testing more tasks, the authors need to significantly scale back their claims about perceptual versus response difficulty. Their results right now merely hint at the possibility that the Yerkes-Dodson effect is based only on perceptual difficulty.

It is interesting to see that at all task levels, an inverted-U is found. Could the claim that the model is reaching arousal states not seen experimentally be verified by comparing the amount of neuromodulation observed experimentally to these gain values?

The authors claim "Since both datasets were curated to be challenging visual search tasks, they required the analysis of complex visual features. Based on this, one would expect an inverted U-shaped gain profile". But their own results suggest there is always an inverted-U, even for easy tasks, if gain is extended long enough. So this finding alone does not strongly support the idea that this is a perceptually challenging task.

What are repetitions/error bars in Figure 5? I also am not clear on what 5C is.

I am curious about which layers contain the most task information relative to each other. It seems as though the middle blocks may be most useful for this average image task, and that could be important for understanding the results and why some layers may be more important than others. It may good to directly determine the layer at which decoding accuracy is highest for each combo of difficulty and gain, in order to explicitly show these trends. At the very least, all panels in 5A (and S2A) should have the same y-axis.

On the whole, I do not understand what the authors believe their scrambling analysis achieves or how to interpret these results. In the Methods they say "we either gradually disrupted later model blocks first and successively added earlier model blocks or the other way around, thereby either leaving mostly earlier or later model blocks intact" Scrambling early blocks does not leave later blocks intact (as their activity relies on the activity patterns in the earlier blocks). Also, keeping the early blocks constant and scrambling later blocks does not allow us to know the impact of early blocks, as that impact is mediated only through the (now scrambled) later blocks (even in the presence of skip connections, the network is still relying on the normal arrangement of the residual inputs to create activity patterns at the next layer; it can't dynamically choose to rely on the skip connections because the residuals are scrambled). I simply don't know if the attempt to ascribe different task performance to different layers is sensible in a hierarchical network. The activity at any given layer only matters for classification to the extent that it can modulate the activity that comes between it and the classifier. In this way the features at different layers aren't separable. I think the decoder analysis is interesting but for similar reasons it also doesn't actually tell us which layers are most important for performance increase at specific gain levels (i.e. what you can readout at a given layer under a given gain condition may not matter if that information has been extensively transformed by the time it actually gets to the classifier layer). It seems like a better framing would be to discuss what effect the early layers have on later layers, such that early layer features can impact performance.

I also don't understand the attempt to relate the amount of gain needed under different scrambling to which layer the network is relying on under different gains in the normal networks. The authors say "Our results so far suggest that high global gain states optimize sensory processing in early blocks. Therefore, we expected that increasingly disrupting late network function (Late to early) and giving more weight to earlier blocks, would lead to a right-ward shift in the performance-gain profiles" You could also expect the opposite: if it is relying more on early layers maybe early layers wouldn't need to be as "loud" as when it's competing with influence from later layers. As far as I can tell there is also a different logic at play here: "Only once also the very last network features were affected, there was a notable increase towards higher gain states for peak performance on the difficult task. This pattern of results suggests that sensory processing in later network blocks is more critical for solving perceptually difficult tasks and that medium global gain states facilitate later processing and hence performance on difficult tasks. " This is saying that if the layer that is important for performance (here, higher layers) is scrambled the network will need higher gain, whereas the first statement seems to say that if the layers *not* needed for performance (also higher layers) are scrambled, the network will need higher gain. The authors need to clarify this seeming contradiction.

The statement "Our results so far suggest that high global gain states optimize sensory processing in early blocks. Therefore, we expected that increasingly disrupting late network function (Late to early) and giving more weight to earlier blocks, would lead to a right-ward shift in the performance-gain profiles" also seems to suggest that you would expect to *not* see such a rightward shift for the difficult task (as Figure 5D shows an optimal gain around only 1 there). Yet the same rightward shift occurs.

I think a more fruitful analysis to understand the role of activity at different layers for easy versus hard tasks may be to simply modulate the gain at different layers individually. It is conceivable, for example, that difficult tasks do rely more on the features at later layers, but the model only needs low to medium gain to activate those layers strongly because the later layers are inheriting impacts from gain changes at earlier layers. Changing gain at individual layers may isolate some of these effects. It may also reveal different behavioral patterns. For example, the authors note that criterion doesn't have a U-shape with gain. In Lindsay & Miller (eLife, 2018) (fig 5E) gain changes (as a model of spatial attention) create a U-shape in criterion when applied at early layers but not at later. Given the sigmoidal activation function used in this model, the propagating impact of gain changes at one layer on later activity and classification is even harder to predict without direct experimentation.

In Discussion:

I'd like to see the authors discuss in more detail (as well as give examples of) what tasks have been considered "easy" versus "difficult" in the experimental literature and how their tasks relate to these concepts as they are used experimentally.

I'd also like to know how using the same gain factor at all layers correspond to biologically data? i.e. does global arousal impact V1-IT to the same extent or are there differences in different areas?

Reviewer #2: The authors examine performance of a deep network on a perceptual categorization task at different levels of difficulty while manipulating the gain of the activation function during testing. Unsurprisingly, they reveal that across a range of tasks, performance on testing is best for a moderate level of gain. However, within a specific task, they also show that increasing the gain can help performance in easier conditions, whereas decreasing the gain can help performance in the most difficult conditions. The authors interpret their results in terms of the Yerkes Dodson relationship between arousal and performance observed in humans.

Overall, I found this paper to be a very fresh take on a longstanding question in cognitive neuroscience – namely how does arousal lead to such a complex pattern of changes in different behavioral tasks. The authors present relatively straightforward simulations in what has now become a relatively well explored “model system” for understanding neural representations, at least in perceptual tasks. However, I have a number of concerns with the manuscript in its current form. Many of them are related to the “middle ground” nature of the model – its not quite such a toy model that its behavior can be easily understood, yet it is missing a lot of features of the real system, and some of those features, such as normalization, might be critically important for the observed results. A complete listing of my concerns is below, and I look forward to seeing them addressed in a revised version of the manuscript.

Major comments:

The task difficulty effects are, in my mind, quite interesting. That said, I’m not sure that the authors have fully explained how they come about – and within this simplified framework, it should certainly be possible. I would venture to guess that the distributions of input magnitude experienced for units at the different levels of difficulty drive the effects – simply put, trials where inputs are a small range of values, or when inputs are all very large, pushing outputs onto the flat part of the activation function – should yield poor performance. So if a given stimulus set (ie. difficult) tends to lead to very high inputs at any cortical layer, then decreasing gain could reduce inputs, and push the input distribution onto the steepest part of the activation function. Is this what is going on here? It seems very testable. For starters the authors could look at overall activity levels for each layer separately for the different difficulty conditions… do they differ? How are they affected by the gain manipulation? Are they just being pushed toward the steep part of the activation function by the best gain? For a given layer, what is the distribution of input received for a given level of difficulty – and where does that fall relative to the activation function? If the activation function from the previous layer were turned down, would that decrease the inputs to this layer and push things onto a steeper part of the activation function? In my mind, the advantage of this model system is that these questions are answerable – and thus it makes sense to see how exactly the effects emerge, rather than speculating a mechanism based on layer specific decoding results.

If all of these effects are driven by changes in the overall activity levels in different layers, it also suggests that these effects should be extremely sensitive to normalization. In the most extreme case, if inputs were normalized to be in the same range for all difficulty levels, then these effects should go away, right? I’m less clear on what happens if outputs are normalized, rather than inputs (ie. divisive normalization), but in the case where these effects are driven by changes in overall activity/input levels, this definitely seems like it should be considered.

A second issue that arises if overall activity changes substantially with different level of gain is that this is, to the best of my knowledge, not what happens in the brain. Instead some neurons increase their activation and others decrease their firing (Devilbiss&waterhouse 2004, Devilbiss&Waterhouse 2011).

Both of these issue could potentially be addressed through a different activity function in which gain is not monotonically linked to overall activity/output.

In general, the results here seem like they would be highly conditioned on the specific details of the activation function. This should be explored and the activation functions used for simulations should be better explained. Based on the figures, the slope of the activation is steepest for small levels of input – but this seems like it would be a poor match to actual cortical pyramidal neurons, which typically require many simultaneous synaptic events to hit spiking threshold. On the other hand, if the key results are robust to the specific sigmoidal function, I would worry less about these details.

In my mind, the fact that a CNN trained using a single gain will show performance that follows an inverse U function of gain, centered on the training gain, is not particularly surprising. This is what is supposed to happen – the model learns weights that are appropriate for trained activation function – if the activation function is changed, performance will suffer, to a degree that depends on the size of the change. In a sense, this is an interesting example of state dependent learning – which actually could be a high-level theory for the Yerkes Dodson inverted U, but probably doesn’t require a sophisticated model. Thus, I see figure 4 as suggesting a limitation of this modeling approach, rather than presenting evidence for the explanatory potential of gain in a feedforward network.

Minor comments:

Figure 6 could use some additional explanation. The x-axis should be clearly explained in the legend. I also was not clear why the authors chose this particular manipulation – rather than one that would be more selective for the gain manipulation (ie. changing gain of one layer at time while leaving others fixed).

The authors motivate work in terms of theories by Aston-Jones and Cohen 2005 – but it also relates to work by Mather 2016, who go into detail about how molecular mechanisms could give rise to a form of gain modulation that increases firing of active neurons, and decreases activation of less active ones (ie. results from devilbiss and waterhouse above).

**Have the authors made all data and (if applicable) computational code underlying the findings in their manuscript fully available?**

Reviewer #1: Yes

Reviewer #2: Yes

PLOS authors have the option to publish the peer review history of their article (what does this mean?). If published, this will include your full peer review and any attached files.

Reviewer #1: No

Reviewer #2: No
---

## [Decision Letter · Decision Letter 1]

1 Feb 2022

Dear Mrs. Sörensen,

Thank you very much for submitting your manuscript "Arousal state affects perceptual decision-making by modulating hierarchical sensory processing in a large-scale visual system model" for consideration at PLOS Computational Biology. As with all papers reviewed by the journal, your manuscript was reviewed by members of the editorial board and by several independent reviewers. The reviewers appreciated the attention to an important topic. Based on the reviews, we are likely to accept this manuscript for publication, providing that you modify the manuscript according to the review recommendations from Reviewer 2.

Sincerely,

Leyla Isik

Associate Editor

PLOS Computational Biology

Thomas Serre

Deputy Editor

PLOS Computational Biology

[LINK]

Reviewer's Responses to Questions

**Comments to the Authors:**

Reviewer #1: I still do not understand the logic of how the scrambling analysis is interpreted, but the authors seem to find it useful.

Reviewer #2: Sorensen and colleagues have revised their manuscript to address the vast majority of my concerns. However, I have a couple of remaining comments:

1) While the ReLu instantiation of the model rules out one possible mechanism through which the observed effects arise, it doesn't answer the larger question of how the effects DO arise in the current setup. I can appreciate that the incremental scrambling approach is intended to get at this question, but at the end of the day I don't find the results of that experiment to provide a satisfying explanation of how the effects come about. To me, a satisfying explanation would require examination of what sorts of information are differentially propagated from layer to layer as you change gain, which almost certainly would require understanding how gain changes in one layer push neural inputs in the next layer towards or away from nonlinearities in the activation function. I think this would be addressable in toy problems. That said, I understand that this may be a question for future work.

2) The long-timescale changes in arousal (as measured with pupil dilation) reported in Vinck, that the authors note relating to higher firing rates in visual regions, are unlikely to be related to norepinephrine signaling, given other work showing that relationship between NE and pupil dilation only holds for high frequency fluctuations (Reimer 2016; Joshi 2016). That said, to the best of my knowledge, the behavioral phenomena that are modeled in the paper have not been explicitly linked to the NE system either. So I would just suggest that the authors use a bit more caution in linking behavioral phenomena to underlying neural systems.

**Have the authors made all data and (if applicable) computational code underlying the findings in their manuscript fully available?**

Reviewer #1: Yes

Reviewer #2: Yes

PLOS authors have the option to publish the peer review history of their article (what does this mean?). If published, this will include your full peer review and any attached files.

Reviewer #1: No

Reviewer #2: No

Figure Files:

Data Requirements:

Reproducibility:

References:

---

## [Editor Report · Decision Letter 2]

26 Feb 2022

Dear Mrs. Sörensen,

We are pleased to inform you that your manuscript 'Arousal state affects perceptual decision-making by modulating hierarchical sensory processing in a large-scale visual system model' has been provisionally accepted for publication in PLOS Computational Biology.

Best regards,

Leyla Isik

Associate Editor

PLOS Computational Biology

Thomas Serre

Deputy Editor

PLOS Computational Biology

---

## [Editor Report · Acceptance letter]

31 Mar 2022

PCOMPBIOL-D-21-01761R2 

Arousal state affects perceptual decision-making by modulating hierarchical sensory processing in a large-scale visual system model

Dear Dr Sörensen,

I am pleased to inform you that your manuscript has been formally accepted for publication in PLOS Computational Biology. Your manuscript is now with our production department and you will be notified of the publication date in due course.

With kind regards,

Livia Horvath
